# Statistical Modelling of Carbonation Process in Reinforced Concrete Structure

**DOI:** 10.3390/ma15082711

**Published:** 2022-04-07

**Authors:** Yinglong Liu, Pengzhen Lin, Zhigang He, Junjun Ma

**Affiliations:** Key Laboratory of Road & Bridge and Underground Engineering of Gansu Province, Lanzhou Jiaotong University, Lanzhou 730070, China; liuyinglong23@163.com (Y.L.); sdohzg945@163.com (Z.H.); majjlz@163.com (J.M.)

**Keywords:** reinforced concrete, carbonation life model, nonlinear statistics, model validation

## Abstract

In order to quantitatively analyze the factors affecting the carbonation of reinforced concrete structures, the carbonation coefficient model is established based on 1834 groups of test data from natural carbonation and indoor accelerated tests in this paper. The main factors considered in the statistical model are the environmental temperature, the concentration of carbon dioxide, relative humidity, water–cement ratio, fly ash replacement, compressive strength of 28 days, curing time, compaction type, exposure to a salt environment, and environmental exposure classes. Based on the multiple nonlinear regression method, the carbonation coefficient model is fitted in two sections according to the different environmental exposures of the concrete structure. To analyze the applicability of the formula, the statistical formulas of relative humidity less than 70% and relative humidity higher than 70% are verified by the test data, and satisfactory results are obtained. Based on the quantitative analysis of the statistical model, the specific effects of relative humidity, strength, carbon dioxide content, fly ash, and curing time on concrete carbonation are clarified. The results show that the factors affecting carbonation are also different with different humidity values in the exposed environment of the concrete structure. When the relative humidity of the exposed environment is less than 70%, the parameters that have a great impact on concrete carbonation are fly ash replacement, compressive strength of 28 days, relative humidity, and the concentration of carbon dioxide. Among them, fly ash replacement, relative humidity, and the concentration of carbon dioxide can promote the carbonation of concrete. When the relative humidity of the exposed environment is higher than 70%, the parameters that have a great impact on concrete carbonation are the concentration of carbon dioxide, relative humidity, compressive strength of 28 days, curing time, and exposure classes. Only the concentration of carbon dioxide is conducive to the carbonation of concrete, and relative humidity has a very significant effect on concrete carbonation.

## 1. Introduction

Due to the long-term resistance to load, physical and chemical erosion, and other deterioration processes of the natural environment, the material properties of reinforced concrete structures inevitably deteriorate prematurely during their service life [1,2]. Carbonation can reduce the PH values of concrete. Once the PH values drop below 9, passivated steel bars will gradually corrode [3,4]. The durability of most concrete is expected to be satisfactory for approximately 50 years [5]. However, the design working life of important buildings such as bridge structures is 100 years or more. It is worrisome that concrete structures in developed countries (as well as in parts of developing countries) have served nearly, or more than, half a century, and some structures may already be in a state of irreparability, with high maintenance costs. It can be predicted that more and more premature durability degradation will occur in such structures, thus bringing about corresponding social and economic consequences.

At present, the design idea of reinforced concrete structure durability under a carbonized environment is as follows: Firstly, we determine the carbonized environment class, then the corresponding limit values of the water–cement ratio, cement content, concrete strength, and thickness of concrete cover are determined. Among them, EN 206-1 [6] defines four classes for carbonation-induced corrosion and gives the maximum water–cement ratio, the minimum cement content, the minimum concrete strength, the minimum thickness of the concrete cover, and other parameters. ACI 318-11 [7] does not make a specific division of the environmental category but rather expounds many factors leading to the deterioration of concrete and measures to prevent harm, and gives the minimum thickness of the concrete cover, maximum water–binder ratio, and minimum concrete strength grade. In addition to specifying the water–cement ratio, cement content, thickness of the concrete cover, and other parameters, the JSCE guidelines for concrete [8] also limit the maximum aggregate size. China’s “Code for Design of Concrete Structures” [9] divides the carbonized environment into three classes and provides design parameters such as the maximum water–binder ratio, minimum concrete strength, and minimum thickness of the concrete cover. It can be seen that the design of reinforced concrete structures’ carbonation durability in various countries is mostly based on qualitative analysis, and it is expected to ensure the durability of reinforced concrete structures by limiting the maximum water–binder ratio (or water–cement ratio), minimum concrete strength grade, minimum cement content, and minimum thickness of the concrete cover, as well as other parameters. The above design methods take the concrete strength, water–binder ratio, and other material parameters and environmental effects as durability design parameters at the same time. However, they are not only repeated but may also cause mutual interference in the value of parameters. Moreover, the functional objectives of each parameter are not clearly distinguished.

Carbonation life analysis models can be divided into two categories: Theoretical model [10] and multi-field coupling numerical model [11]. Among them, the initial theoretical model sets up the modeling of the diffusion process of carbon dioxide based on Fick’s first law [12]. The subsequently developed theoretical models consider certain factors that influence carbonation based on this formula and modify the carbonation coefficient [3,13,14,15,16,17]. The difficulty of applying this model is the estimation of the carbonation coefficient. Although the multi-field coupling numerical model can comprehensively reflect the influence of various factors on carbonation, it needs to be discretized in the space domain and time domain and involves the solution of nonlinear partial differential equations, which requires a large amount of calculation and is inconvenient for application. Although the theoretical model has some defects, its structure is simple and easy to use, and satisfactory results have been obtained under stable environmental conditions such as laboratory conditions. Therefore, it is still the most widely used model at present. When applying the theoretical formula to carbonation analysis, there is no consensus on how to determine the carbonation coefficient, which mainly reflects the durability index of concrete and needs to consider the content of carbon dioxide, strength, air humidity, temperature, concrete exposure conditions, and other factors.

In order to quantitatively analyze the factors affecting carbonation, this paper studied a large number of papers related to carbonation and selected relatively complete test data of durability parameters. Among them, the carbonation test data include 1834 samples [14,15,16,17,18,19,20,21,22,23,24,25,26,27,28,29,30,31,32,33,34,35,36,37]. The carbonation coefficient model was obtained based on nonlinear statistical analysis.

## 2. Methods

The carbonation coefficient is the dependent variable when using multivariate nonlinear regression to analyze the carbonation coefficient. Then, considering factors affecting concrete’s durability and referring to the research results within the literature [3,14,15,16,17,18,19,20,21,22,23,24,25,26,27,28,29,30,31,32,33,34,35,36,37], the independent variables are mainly as follows: The environmental temperature (for ease of expression, this is simplified to T and the expression of subsequent parameters is the same), the concentration of carbon dioxide (*C*), relative humidity (*RH*), water–cement ratio (*W/C*), fly ash replacement (*FA*), 28-day compressive strength (*f_c_*), curing time (*D*), compaction type, exposure to a salt environment, environmental exposure classes (*X*), and whether there are waterproof measures.

For the above independent variables, considering the needs of modeling, some independent variables can only be used after coding, except the data for temperature and carbon dioxide concentration. Normal compaction and self-compacting concrete are coded as 1 and −1, respectively. If the exposed concrete has rain protection measures, the code is 1; otherwise, it is −1. Four classes for carbonation-induced corrosion of EN206-1 [6] are shown in Table 1. In this order, the environmental exposure classes are coded as 1–4 in the modeling.

The coefficient of concrete carbonation is analyzed by the stepwise regression method in SPSS software. The advantage of this method is conducting a simple regression of all explanatory variables using the explained variables first, then taking the regression equation with the largest contribution to the explanatory variables as the basis, and gradually introducing other explanatory variables. Finally, the explanatory variables retained in the model are not only important but also have no serious multicollinearity.

In this paper, 1834 samples [14,15,16,17,18,19,20,21,22,23,24,25,26,27,28,29,30,31,32,33,34,35,36,37] of carbonation test data were selected, among which 568 samples were obtained from accelerated carbonation tests. For concrete structures exposed to natural conditions, if not described in the literature, carbon dioxide was considered 0.03%, and the temperature and relative humidity were considered the average value of the environment where the structure is located.

In all selected samples, the lowest carbon dioxide concentration was 0.02% and the highest was 45% after the outliers were removed. The relative humidity was between 40% and 85%, the water–cement ratio ranged from 0.31 to 1.9, and the lowest and highest 28-day compressive strength was 9.3 Mpa and 84.3 Mpa, respectively. The maximum content of fly ash was 70%. Some concrete had rain-protection measures. The shortest and longest curing time of concrete was 1 day and 191 days, respectively. The coding range of environmental effect classes was 1 to 4.

Due to the research content of each paper being different, some information may be missing when the complete experimental data are used for another study. In order to maximize the accuracy of the regression model, it is necessary to process the missing data. There are a number of methods to process missing data, one of which is to eliminate all cases of missing data, with the advantage that all data are the original data, and if the proportion of the missing values is low, the method will be very effective [38]. The disadvantage is that in order to obtain complete information by reducing the specimen size, a large amount of hidden information may be lost, which may result in deviation of the data, leading to an erroneous conclusion if too many data are removed. When the amount of missing data is large, the case exclusion method will no longer be applicable, and the mean value of all data can be filled by the mean substitution method. This method will not affect the mean value estimation of variables, but the mean substitution method assumes data are missing completely at random and will lead to a decrease in the variance and standard deviation of variables. Other complex interpolation methods may achieve better results in terms of statistical significance, but regression analysis for the carbonation coefficient may distort the data, causing result error. In this paper, samples with complete data were selected as much as possible. In total, 78 of 1834 data were incomplete, accounting for approximately 4.25% of the entire sample, and mean substitution was adopted to interpolate the missing data. In order to ensure the authenticity and reasonableness of statistical data, it is necessary to delete the data with obvious anomalies. Data where the deviation between the standard deviation and the average value exceeds 3 are considered abnormal and need to be deleted, accounting for 15.27% of the specimens.

## 3. Results and Discussion

*RH* will affect the degree of capillary saturation of reinforced concrete structures, and then affect the carbonation rate of concrete [39]. Concrete pore water is the place where carbonation occurs, and the formation of its liquid environment is directly related to the relative humidity of concrete exposure conditions. If *RH* in the exposed environment is comparatively low, the saturation of pore fluid in the structure is correspondingly low as well. Therefore, the diffusion rate of *C* is high. However, the carbonation rate slows down due to the lack of required moisture. If *RH* is comparatively high, the pore fluid with high saturation hinders the diffusion of CO_2_ and slows down the carbonation rate of concrete [40]. Bakker [41] found that concrete would hardly be carbonized in a saturated solution. Y. Lo [16] believes that the optimal humidity for concrete carbonation is 50–70%. Therefore, theoretically, RH has an extreme point that can cause the carbonation rate of concrete to reach the maximum. In order to obtain a more accurate concrete carbonation coefficient estimation model, statistical analysis of the carbonation coefficient was divided into two sections according to *RH* in this paper. Concrete exposed to less than 70% RH is in the carbonation growth point; conversely, it is classified in the downward zone.

There are 1162 groups of samples with RH ≤ 70%. Three groups of data are incomplete after excluding 220 samples of abnormal data. For samples with RH > 70%, 30 groups of data are incomplete after excluding 60 samples of abnormal data. All incomplete data are processed by the mean imputation method.

### 3.1. The Carbonation Coefficient Estimation Model with RH Less than 70%

The statistical characterization of the variables of the concrete carbonation coefficient when *RH* is less than 70% is shown in Table 2.

The fitting degree of each independent and dependent variable was analyzed through curve estimation, and the power of corresponding independent variables in the carbonation coefficient model was determined according to the curve shape that generated the maximum *R*^2^. The results show the variables that have a relatively great impact on the concrete carbonation coefficient are *C*, *f*_c_, *RH*, *D*, X, *T,* and *FA*. The specific correlations between each parameter and the carbonation coefficient are shown in Figure 1. Among them, Figure 1a–g are the maximum curve estimation of *RH*, *f*_c_, *T*, *C*, *D*, *FA* and X and carbonation coefficient respectively.

It can be seen from Figure 1 that when *RH* is less than 70%, the parameters with great correlations with the carbonation coefficient (*R*^2^ > 0.4) are *FA*, C, *f*_c_, and *RH,* respectively. Therefore, the carbonation coefficient can be expressed as:(1)k=a.EXP(b.FA2).c.(C)0.3.(dfc).e.(RH0.7)0.2

The starting values of the parameters are *a* = 40, *b* = *c* = *d* = *e* = 2. The parameter constraint is *a* > 0, *b* > 0, *c* > 0, *d* > 0, *e* > 0.

The parameter estimates and ANOVA table of the regression model are shown in Table 3 and Table 4, respectively.

It can be seen from Table 4 that the *R*^2^ value of the estimation model is 0.858, which indicates that 85.8% of the influencing factors of the specimen’s carbonation coefficient can be explained by this model, which is statistically significant. Therefore, when *RH* is less than 70%, the estimated model of the carbonation coefficient is:(2)k=41.94EXP(2.24FA2)2.05(C)0.3(1.02fc)2.3(RH0.7)0.2

### 3.2. The Carbonation Coefficient Estimation Model with RH Higher than 70%

The variable eigenvalues of the carbonation coefficient when *RH* is higher than 70% are shown in Table 5.

When *RH* is higher than 70%, it can be seen from Figure 1 that the parameters with great correlations with the carbonation coefficient (*R*^2^ > 0.4) are *C*, *f*_c,_ RH, *D,* and *X,* respectively.

Therefore, the carbonation coefficient can be expressed as:(3)$$k=a{\left(C/0.03\right)}^{0.4}b{\left(RH/0.7\right)}^{-0.2}cf_{c}^{-1.0}dD^{-0.2}eX^{-0.2}$$

The starting values of the parameters are *a* = 40, *b* = *c* = *d* = *e* = 1.5. The parameter constraint is *a* > 0, *b* > 0, *c* > 0, *d* > 0, *e* > 0.

The parameter estimates and ANOVA table of the regression model are shown in Table 6 and Table 7, respectively.

Table 7 shows that the *R*^2^ value of the estimated model is 0.972, indicating that the specimen’s carbonation coefficient is highly correlated with the calculated results of the model. Therefore, when the relative humidity is higher than 70%, the carbonation coefficient estimation model is:(4)k=38(C0.03)0.41.5(RH0.7)−4(1.38fc)1.5D−0.21.5X−0.2

Through regression analysis, the concrete carbonation coefficient estimation model can be expressed as:(5){k=41.94EXP(2.24FA2)2.05(C)0.3(1.02fc)2.3(RH0.7)0.2    RH≤70%k=38(C0.03)0.41.5(RH0.7)−4(1.38fc)1.5D−0.21.5X−0.2      RH>70%

### 3.3. Formula Verification

In order to verify the applicability of the formula, the carbonation estimation results based on the statistical formula are compared with the test values. One part of the concrete carbonation test data is the indoor accelerated test of the concrete box girder model, and the other part is from the concrete box girder bridge exposed to nature.

Among them, the 28-day measured compressive strength of concrete used for indoor accelerated carbonation is 49.6 MPa. The specific size, mix proportion, and material composition of the concrete box girder model can be found in the literature [42].

Five concrete box girder models were constructed and carbonized for 40 days in a carbonation box with *RH*, *C,* and *T* of 70%, 20%, and 20 ± 2 °C, respectively.

When testing the carbonation data, the five box girders were numbered 1#~5#, and phenolphthalein was used to measure the carbonation depth. Among them, 100 groups of carbonation data were tested on each roof of the model box girder. The carbonation data of the left and right webs and floor were from 20 groups and 40 groups, respectively. In order to eliminate error as much as possible, the carbonation data of the roof, floor, and web were averaged. For example, the carbonation depth of the roof was the average of 100 groups of the carbonation depth.

The concrete bridge exposed to the natural environment is a highway three-span continuous beam. The bridge is located in the reservoir area, and the exposure environment belongs to class 4.

The curing time of the beam was 21 days. It was exposed to the natural environment for 10 years during the test. The strength tested by the rebound instrument was 51 MPa. The carbon dioxide content in the atmosphere was considered 0.03%. The annual average *RH* and *T* of the bridge site were 75% and 9.3 °C, respectively.

The carbonation depth of the web and floor of the left midspan (for ease of expression, this is simplified to LMD, and the expression of subsequent parameters is the same), right midspan (RMD), and midspan (MD) sections were tested.

Among them, five groups of carbonation data were tested on both sides of the web and seven groups of carbonation data were tested on the floor. Similarly, the mean carbonation value was used as the carbonation data of the corresponding plate. 

The measured carbonation results of the samples and the calculated values of the statistical model are shown in Table 8.

It can be seen from Table 1 that for the carbonation environment with *RH* ≤ 70%, except for the predicted value of the 4# beam being small, the other values are generally close to the measured values. For the box girder exposed to the natural environment, there is little difference between the predicted value of the model and the measured value, except for individual measuring points.

For the convenience of analysis, the difference rate *ƞ* is introduced, and the expression is:(6)$$\eta =\frac{Test\;value-Formula\;calculated\;value}{Test\;value}\times 100\%$$

In order to make all the differences positive, all the differences calculated by each model are taken as absolute values. The difference between the predicted value and the measured value of the formula proposed in this paper is shown in Figure 2.

It can be seen from Figure 2 that when *RH* ≤ 70%, the maximum difference between the predicted value of the model and the measured value is 14.2%, and the difference is generally within 10%. When *RH* > 70%, for the box girder in the natural environment, the difference between the midspan floor and the web of the right span is less than 2%, and the difference between the left web and the left span is the largest, which is 16.7%. The difference of other measuring points is within 10%.

In summary, the overall difference of the model proposed in this paper is small. Considering the discreteness of the test data it is considered that the carbonation statistical model proposed in this paper has good accuracy and can be used to predict the carbonation life of concrete.

### 3.4. Discussion

In order to quantitatively analyze the effect of each variable on concrete carbonation, the concrete carbonation contribution coefficient *m_i_* is introduced, where *m_i_* is the coefficient of each variable in the carbonation coefficient (*k*) of concrete. Taking *RH* as an example, the concrete carbonation contribution coefficient *m_i_* of this variable can be expressed as:(7){mi=(RH0.7)0.2       RH≤70%mi=(RH0.7)−0.2    RH>70%

The influence curves of *m_i_* of *FA, RH, fc, C, D* and *X* are shown in Figure 3, and as can be observed, *FA*, *RH*, *f_c_*, *C*, *D**,* and *X* can all have a significant effect on concrete carbonation. Among them, Figure 3a–f are the relationship between *FA*, *RH*, *f_c_*, *C*, *D*, *X* and carbonation contribution coefficient respectively.

From Figure 3a, it can be seen that the influence curve of *FA* content on the carbonation coefficient is an exponential distribution. When *RH* ≤ 70%, the carbonation coefficient increases gradually with the increase in fly ash content. When the content of *FA* is less than 20%, fly ash has almost no effect on concrete carbonation. Under the same environmental conditions, the carbonation depth of concrete with 60% *FA* content is about 2 times of that of 20%. As shown in Figure 3b, when *RH* is less than 70%, the carbonation rate of concrete increases gradually with the increase in *RH*. When *RH* exceeds 70%, *RH* will significantly inhibit the carbonation of concrete. When *RH* increases from 70% to 90%, the carbonation rate of concrete will be reduced by about 4.2 times, and this conclusion is consistent with the literature [13]. The most favorable *RH* for concrete carbonation is between 56 and 70%, which is consistent with the conclusions within the literature [16,40]. In Figure 3c, the value of *f_c_* in two *RH* environments has the same inhibitory effect on carbonation, and the carbonation depth of concrete with an *f_c_* value of 10 Mpa is about 5.6 times of that of an *f_c_* value of 65 MPa. The value of *f_c_* has an obvious inhibitory effect on the carbonation of concrete. This conclusion is consistent with that of the existing literature. In terms of *C*, it can be seen from Figure 3d that when *RH* is higher than 70%, the promotion effect of *C* on carbonation is more significant. As is shown in Figure 3e, the *D* value of concrete has an inhibition effect on carbonation. However, when the *D* value of concrete is more than 28 days, the inhibition of carbonation will be very small. It can be seen from Figure 3f that the carbonation rate of concrete decreases with an increasing environmental *X* value. When *RH* > 70%, the carbonation rate is more sensitive to the value of *X*. When the value of *X* increased from 1 to 4, the carbonation rate only decreased by about 1.3 times. The inhibition effect of *X* was not obvious, which may be related to the smaller number of samples.

## 4. Conclusions

In order to explain the concrete carbonation of concrete structures, 1834 samples of experimental data are selected from the existing relevant literature, and the statistical model of the carbonation coefficient is established by the multivariate nonlinear statistical method. The applicability of the model is verified by indoor and outdoor measured data. The results show that the carbonation statistical model proposed in this paper is robust. Through the quantitative analysis of various parameters in the statistical model, the influence of concrete materials and environmental conditions on carbonation is clarified. The main conclusions are as follows:(1)*RH* has two effects on the carbonation of concrete. When *RH* is less than 70%, the increase in *RH* will promote carbonation. If it exceeds 70%, the carbonation of concrete will be restrained. When *RH* increases from 70% to 90%, the carbonation rate of concrete will be reduced by about 4.2 times.(2)When *RH* is less than 70%, the main parameters that have a great impact on concrete carbonation are *FA*, *RH*, *f_c_,* and *C*. Among them, only *f_c_* can inhibit carbonation. *RH* and *C* can greatly promote concrete carbonation. When the content of *FA* is less than 20%, it has little effect on the carbonation of concrete. However, when the content exceeds 20%, it will significantly promote carbonation. Therefore, in order to ensure the carbonation durability of concrete, the content of *FA* is recommended to be controlled within 20%.(3)When *RH* is higher than 70%, *RH*, *f_c_, D,* and *C* have a great impact on the carbonation of concrete. Only *C* can promote carbonation, and the other factors are conducive to the carbonation durability of concrete. Among them, if *D* is within 28 days, it will have a significant correlation with carbonation. The curing time of concrete should be increased as much as possible.

It should be noted that the carbonation statistical model in this paper is suitable for concrete members without stress. Follow-up studies will increase the sample size and further consider the stress effect, so as to obtain a more complete concrete carbonation model.

## Figures and Tables

**Figure 1 materials-15-02711-f001:**
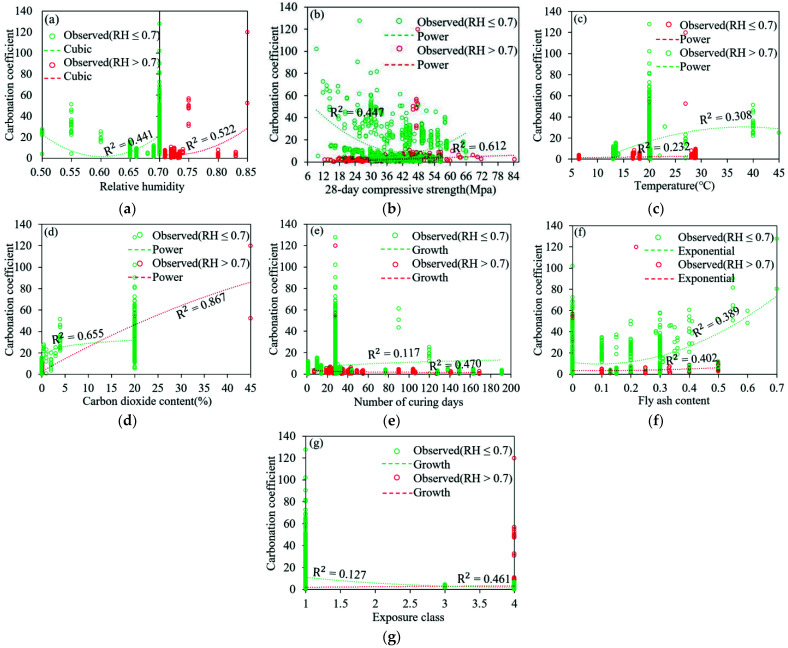
Correlation between each parameter and carbonation coefficient (**a**–**g**) are the maximum curve estimation of *RH*, *f*_c_, *T*, *C*, *D*, *FA* and X and carbonation coefficient respectively.

**Figure 2 materials-15-02711-f002:**
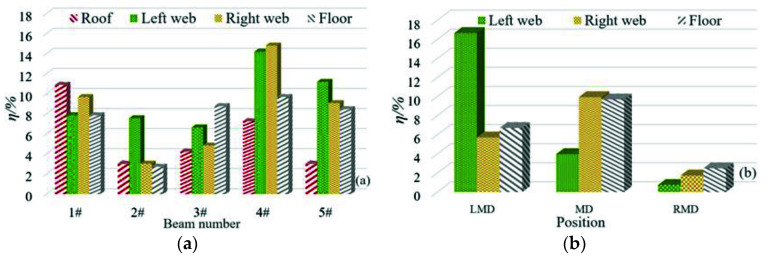
*ƞ* of chloride concentration at each measuring point of the test box girder; (**a**) *ƞ* of *RH* less than 70; (**b**) *ƞ* of *RH* higher than 70.

**Figure 3 materials-15-02711-f003:**
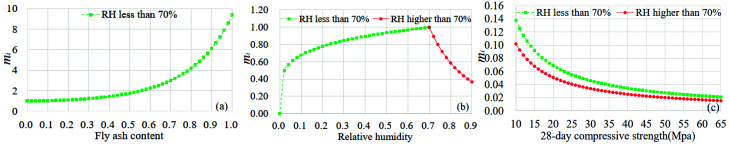
Influence curves of *m_i_* of each variable on carbonation coefficient (**a**–**f**) are the relationship between *FA*, *RH*, *f_c_*, *C*, *D*, *X* and carbonation contribution coefficient respectively.

**Table 1 materials-15-02711-t001:** Classes for carbonation-induced corrosion according to EN206-1 [6].

Class	Environment Description
1	Dry or permanently humid
2	Humid, rarely dry
3	Moderately humid
4	Cyclically humid and dry

**Table 2 materials-15-02711-t002:** Characterization of the variables with RH less than 70% after removing the outliers.

Numerical Values	Average Value	Rage of Results
Carbon dioxide content	5.72%	0.02–20%
Relative humidity	67.24%	40–70%
Temperature	17.98 °C	13 °C–45 °C
Water/cement ratio	0.52	0.35–0.96
28-day compressive strength	38.81 Mpa	9.3–66
Number of curing days	31.15 days	1 day–191 days
Fly ash content	0.10	0%~70%
Compaction type	1	−1~1
Exposure to salts	−1	−1~1
Protection from the action of rain	−1	−1~1
Exposure class	2	1~4

**Table 3 materials-15-02711-t003:** Parameter estimates with *RH* less than 70%.

Parameter	Estimate	Standard Error	95% Confidence Interval	
			Lower Bound	Upper Bound
*a*	41.94	43.36	−79.90	60.25
*b*	2.24	0.097	2.065	2.444
*c*	2.05	194.38	−5.07	7.02
*d*	1.02	160.30	−88.48	90.47
*e*	2.30	171.21	−44.93	46.91

**Table 4 materials-15-02711-t004:** ANOVA table with *RH* less than 70%.

Source	Sum of Squares	*df*	Mean Squares
Regression	351,789.838	8	43,973.730
Residual	32,470.660	927	35.028
Uncorrected total	384,260.498	935	-
Corrected total	227,874.508	934	-

Dependent variable: *k*. *R*^2^ = 1 − (Residual sum of squares) / (Corrected sum of squares) = 0.858.

**Table 5 materials-15-02711-t005:** Parameter estimates with *RH* higher than 70% after removing the outliers.

Numerical Values	Average Value	Rage of Results
Carbon dioxide content	0.50%	0.03–45%
Relative humidity	73.01%	71–85%
Temperature	20.61 °C	6.4–28.9 °C
Water/cement ratio	0.54	0.31–1.9
28-day compressive strength	37.21 Mpa	12.5–84.3
Number of curing days	37.21 days	7–169 days
Fly ash content	0.12	0–50%
Compaction method	1	−1–1
Exposure to salts	−1	−1–1
Protection from the action of rain	−1	−1–1
Exposure class	2	1–4

**Table 6 materials-15-02711-t006:** Parameter estimates with *RH* higher than 70%.

Parameter	Estimate	Standard Error	95% Confidence Interval	
			Lower Bound	Upper Bound
*a*	38.00	16.91	−75.31	90.12
*b*	1.50	0.076	1.20	1.50
*c*	1.38	61.08	−54.51	56.52
*d*	1.50	62.39	−66.54	68.39
*e*	1.50	64.24	−76.64	78.771

**Table 7 materials-15-02711-t007:** ANOVA table with *RH* higher than 70%.

Source	Sum of Squares	*df*	Mean Squares
Regression	39,529.737	8	4941.217
Residual	903.623	561	1.611
Uncorrected total	40,433.361	569	-
Corrected total	33,013.163	568	-

Dependent variable: *k*. *R*^2^ = 1 − (Residual sum of squares) / (corrected sum of squares) = 0.972.

**Table 8 materials-15-02711-t008:** Measured value and model calculated value.

Structural Part	Beam Number	*RH* ≤ 70 (mm)	Model Value (mm)	Position	*RH* > 70 (mm)	Model Value (mm)
Roof	1#	2.95	3.31	LMD	—	3.43
	2#	3.41				
	3#	3.17		MD	—	
	4#	3.55				
	5#	3.21		RMD	—	
Left web	1#	3.05	3.31	LMD	3.35	3.43
	2#	3.56				
	3#	3.09		MD	3.86	
	4#	3.78				
	5#	3.68		RMD	4.05	
Right web	1#	2.99	3.31	LMD	3.79	3.43
	2#	3.41				
	3#	3.15		MD	4.46	
	4#	3.80		RMD	3.95	
	5#	3.61				
Floor	1#	3.05	3.31	LMD	3.75	3.43
	2#	3.22				
	3#	3.02		MD	4.52	
	4#	3.63				
	5#	3.59		RMD	4.12	

## Data Availability

The data used to support the findings of this study are available from the corresponding author upon request.

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
