# Peer review of "Statistical Modelling of Carbonation Process in Reinforced Concrete Structure"

_materials, 2022, doi:10.3390/ma15082711_

Round 1
Reviewer 1 Report
The paper topic is related to the establishment of a carbonation coefficient model based on data collected in the literature. The paper has novelty and is focused on an interesting subject and is important nowadays.
For its publication some issues need to be explained:
i) In chapter 2 - methods, pg 3, is referred that it was chosen for the analysed 1834 group, but it is not explained how it was done this selection (methodology used, criteria considered and not considered);
ii) Figure 1, R2 are very low with the exception of CO2, so how can be possible to make conclusions about these fittings, as those presented in the paragraph before the figure.
iii) in the paragraph before equation 1, is said that R2=4, how the value was achieved?
iv) minor detail, but the Table 7 R2 is 0.972, not 0.973
v) another minor detail in chapter 3.4, put all "mi" in italic format.
Author Response
Dear Reviewer:
  Thank you for your letter and for the reviewers’ comments concerning our manuscript entitled Statistical modelling of Carbonation Life in Reinforced Concrete Structure. Those comments are all valuable and very helpful for revising and improving our manuscript, as well as the important guiding significance to our researches. We have studied comments carefully and have made correction which we hope meet with approval. The main corrections in the manuscript and the responds to the reviewer’s comments are as following:
comments:
  i) In chapter 2-methods, pg 3, is referred that it was chosen for the analysed 1834 group, but it is not explained how it was done this selection (methodology used, criteria considered and not considered);
- ii) Figure 1, R2 are very low with the exception of CO2, so how can be possible to make conclusions about these fittings, as those presented in the paragraph before the figure.
iii) in the paragraph before equation 1, is said that R2=4, how the value was achieved?
- iv) minor detail, but the Table 7 R2 is 0.972, not 0.973
- v) another minor detail in chapter 3.4, put all "mi" in italic format.
 Response to reviewer:
  i) We are very sorry for our unclear report. The basis of data selection is to ensure that all parameters of statistical analysis in this paper are not missing as far as possible. However, considering the long carbonation time and the possible loss of temperature and carbon dioxide content in natural exposure environment, this paper is considered as the mean value and has been supplemented in the revised draft.
- ii) In curve estimation, the figure shows the corresponding power when each parameter is most related to the carbonation coefficient. The R2 of carbon dioxide content is large because the carbon dioxide content has the greatest impact on the carbonation coefficient. Through curve estimation, when the curve of 0.3 power is taken, the maximum R2 can be provided. If R2 of other parameters is small, it indicates that the influence on carbonation coefficient is not the largest.
iii)We are very sorry for the mistake in the original paper. The actual value is 0.4. Considering the actual curve estimation results, when R2 exceeds 0.4, this paper believes that the parameter will have a great impact on the carbonation coefficient.
iv), v) These errors have been corrected in the revised paper.
Special thanks to you for your good comments.

Reviewer 2 Report
The article is about statistical modelling of carbonation life in reinforced con-crete structure. However, some issues must to be addressed:
- Abstract: Please start by expressing the aim of this paper, followed by the rest of the information. Also, please define or try to avoid using abbreviations in the abstract. Typically, the abstract should provide a broad overview of the entire project, summarize the results, and present the implications of the research or what it adds to its field.
- Please avoid bulk citation, like [37-39], [41-45], [10-33].
- Please improve introduction section by adding information from newest publications in the field, added to section references (2021-2022).
- How was obtained eq. (1) or (2)?
- Figure 2: seems to not have scientific meaning, pleas remove it.
- The results are merely presented, not properly discussed. Please add explanations for the observed changes. Please give an extended discussion on the obtained results and correlate your findings with previous literature studies and prospective applications.
- The authors must to provide some details about importance of the research and their applicability.
- Please enhance the clarity of the conclusion section in order to highlight the results obtained.
- General check-up and correction of the English language is suggested. There are still some minor typos and grammatical errors.
The author needs to address the abovementioned points for the betterment of the manuscript.
Author Response
Dear Reviewers
  Thank you for your letter and for the reviewers’ comments concerning our manuscript entitled Statistical modelling of Carbonation Life in Reinforced Concrete Structure. Those comments are all valuable and very helpful for revising and improving our manuscript, as well as the important guiding significance to our researches. We have studied comments carefully and have made correction which we hope meet with approval. The main corrections in the manuscript and the responds to the reviewer’s comments are as following:
comments:
- Abstract: Please start by expressing the aim of this paper, followed by the rest of the information. Also, please define or try to avoid using abbreviations in the abstract. Typically, the abstract should provide a broad overview of the entire project, summarize the results, and present the implications of the research or what it adds to its field.
- Please avoid bulk citation, like [37-39], [41-45], [10-33].
- Please improve introduction section by adding information from newest publications in the field, added to section references (2021-2022).
- How was obtained eq. (1) or (2)?
- Figure 2: seems to not have scientific meaning, pleas remove it.
- The results are merely presented, not properly discussed. Please add explanations for the observed changes. Please give an extended discussion on the obtained results and correlate your findings with previous literature studies and prospective applications.
- The authors must to provide some details about importance of the research and their applicability.
- Please enhance the clarity of the conclusion section in order to highlight the results obtained.
- General check-up and correction of the English language is suggested. There are still some minor typos and grammatical errors.
 Response to reviewer:
- We are very sorry for our unclear report. The abstract of the revised draft has been restated.
2,3. These questions have been modified.
- According to the curve estimation in Fig. 1, based on the maximum R2 provided by the parameter curve for the carbonation coefficient, the power of the corresponding parameters in the formula is given. For example, when estimating the curve, if the power of carbon dioxide content is 0.3(if RH≤70%), it will be most related to the carbonation coefficient, so as to obtain the maximum R2.
- It has been deleted in the revised paper.
- Relevant descriptions have been added in the revised paper.
- It has been provided in the revised paper.
- The conclusion in the revised paper has been restated.
- The revised manuscript has been carefully checked and corrected.
Special thanks to you for your good comments.

Reviewer 3 Report
the file is attached

Author Response
Dear Reviewer:
  Thank you for your letter and for the reviewers’ comments concerning our manuscript entitled Statistical modelling of Carbonation Process in Reinforced Concrete Structure. Those comments are all valuable and very helpful for revising and improving our manuscript, as well as the important guiding significance to our researches. We have studied comments carefully and have made correction which we hope meet with approval. The main corrections in the manuscript and the responds to the reviewer’s comments have been replied and marked in the attached paper.

Round 2
Reviewer 2 Report
The article is suitable for publication.